# RNA-Guided *As*Cas12a- and *Sp*Cas9-Catalyzed Knockout and Homology Directed Repair of the Omega-1 Locus of the Human Blood Fluke, *Schistosoma mansoni*

**DOI:** 10.3390/ijms23020631

**Published:** 2022-01-06

**Authors:** Wannaporn Ittiprasert, Chawalit Chatupheeraphat, Victoria H. Mann, Wenhui Li, André Miller, Taiwo Ogunbayo, Kenny Tran, Yousef N. Alrefaei, Margaret Mentink-Kane, Paul J. Brindley

**Affiliations:** 1Department of Microbiology, Immunology & Tropical Medicine, & Research Center for Neglected Diseases of Poverty, School of Medicine & Health Sciences, George Washington University, Washington, DC 20037, USA; chawalit.cha@mahidol.ac.th (C.C.); vmann@gwu.edu (V.H.M.); lwh0317@163.com (W.L.); yalrefaei@me.com (Y.N.A.); 2Center for Research and Innovation, Faculty of Medical Technology, Mahidol University, Salaya, Nakhon Pathom 73170, Thailand; 3Lanzhou Veterinary Research Institute, Chinese Academy of Agricultural Sciences, Lanzhou 730050, China; 4Schistosomiasis Resource Center, Biomedical Research Institute, Rockville, MD 20850, USA; amiller@afbr-bri.org (A.M.); ogunbayo@gmail.com (T.O.); kennyt2012@gmail.com (K.T.); mmentinkkane@afbr-bri.org (M.M.-K.); 5Department of Medical Laboratory Technology, College of Health Sciences, PAEET, Adailiya, Kuwait City 73101, Kuwait

**Keywords:** *As*Cas12a, *Sp*Cas9, ribonucleoprotein complex, CRISPR, omega-1, genome editing, homology directed repair, nonhomologous end-joining, schistosome egg

## Abstract

The efficiency of the RNA-guided *As*Cas12a nuclease of *Acidaminococcus* sp. was compared with *Sp*Cas9 from *Streptococcus pyogenes*, for functional genomics in *Schistosoma mansoni*. We deployed optimized conditions for the ratio of guide RNAs to the nuclease, donor templates, and electroporation parameters, to target a key schistosome enzyme termed omega-1. Programmed cleavages catalyzed by Cas12a and Cas9 resulted in staggered- and blunt-ended strand breaks, respectively. *As*Cas12a was more efficient than *Sp*Cas9 for gene knockout, as determined by TIDE analysis. CRISPResso2 analysis confirmed that most mutations were deletions. Knockout efficiency of both nucleases markedly increased in the presence of single-stranded oligodeoxynucleotide (ssODN) template. With *As*Cas12a, ssODNs representative of both the non-CRISPR target (NT) and target (T) strands were tested, resulting in KO efficiencies of 15.67, 28.71, and 21.43% in the *Sp*Cas9 plus ssODN, *As*Cas12a plus NT-ssODN, and *As*Cas12a plus T-ssODN groups, respectively. *Trans*-cleavage against the ssODNs by activated *As*Cas12a was not apparent in vitro. *Sp*Cas9 catalyzed more precise transgene insertion, with knock-in efficiencies of 17.07% for the KI_Cas9 group, 14.58% for KI_Cas12a-NT-ssODN, and 12.37% for KI_Cas12a-T-ssODN. Although *As*Cas12a induced fewer mutations per genome than *Sp*Cas9, the phenotypic impact on transcription and expression of omega-1 was similar for both nucleases.

## 1. Introduction

Interest is increasing in functional genomics to investigate helminth parasites, specifically genome editing to investigate pathophysiology, carcinogenesis, and biology. Recent progress showcased the deployment of clustered regularly interspaced short palindromic repeats CRISPR/Cas-programmed genome editing in parasitic flatworms, in particular with the human blood fluke *Schistosoma mansoni* [1] and the carcinogenic liver fluke, *Opisthorchis viverrini* [2] as exemplars. However, given that these helminth parasites are nonmodel organisms, that their genome sequences are often not available or not well curated, and that they are frequently refractory to maintenance in the laboratory, approaches to advance functional genomics using CRISPR/Cas are not yet established or available for wide use in the field of tropical medical and neglected tropical diseases. Nonetheless, our capacity to decipher the molecular pathogenesis of these diseases will be aided by advances in gene editing methodologies [3].

RNA-guided nucleases evolved in archaea and bacteria as adaptive immunity mechanism to destroy phage viruses and other nucleic acid invaders [4,5]. The catalogue of these CRISPR/Cas nucleases is expanding. The *Sp*Cas9 of *Streptococcus pyogenes*, a Class 2, type II Cas enzyme in the phylogeny of Koonin and coworkers [5], dominates diverse gene-editing applications in biomedicine and beyond [6,7,8], given its programmable, precise double-stranded DNA (dsDNA) cleavage, accomplished by two catalytic domains, RuvC and HNH, of the nuclease at the target site complementary to the guide RNA [9]. A second Class 2 family, the type V nucleases, as represented by Cas12a (formerly known as Cpf1), uses a single RuvC catalytic domain for guide RNA-directed dsDNA cleavage [10,11]. Unlike *Sp*Cas9, Cas12a orthologues recognize a T nucleotide-rich protospacer-adjacent motif (PAM), catalyze the maturation of their own guide CRISPR RNA (crRNA), and induce a dsDNA break, distal to the PAM, bearing staggered 5′- and 3′-ends [10]. These Cas12a features are also attractive for complex genome editing, including in plants [12]. For their maturation, the crRNAs of type II CRISPR-Cas systems require processing by a cognate *trans*-activating CRISPR RNA [13]. Moreover, the Cas12a nuclease is smaller in size than *Sp*Cas9 and uses a shorter CRISPR RNA (crRNA) for activity [14]. Notably, target site-activated Cas12a exhibits nonspecific *trans* activity against bystander nucleic acids [15,16], an attribute that has been harnessed in a burgeoning range of biosensors [17,18].

Delivery of the CRISPR system reagents into the cells of the multicellular parasite *S. mansoni* in the form of ribonucleoprotein (RNP) complexes may enable immediate and permanent programmed gene editing and obviate concerns with CRISPR reagent longevity or integration in the host cell genome as lentiviral system [1]. Programmed CRISPR/Cas editing using RNP-based delivery can precede analysis of mutated target regions that rely on cloning and CRISPR vector construction, as addressed in our earlier reports [1,2]. Previously, we deployed both RNP and lentiviral-delivered CRISPR/*Sp*Cas9 approaches to knockout (KO) and knock-in (KI) into a blunt end at *Sp*Cas9 cleavage site at the multiple copies of the omega-1 (*ω*1) gene in parasite eggs. To advance functional genomics for schistosomes and aiming to improve the mutagenesis and transgene knock-in of CRISPR efficiency in schistosomes, which exhibit AT-rich genomes, we investigated the nuclease activity of *Acidaminococcus* sp. Cas12a (*As*Cas12a) in comparison with *Sp*Cas9. These CRISPR systems differ in key ways: first, *Sp*Cas9 recognizes NGG as the protospacer-adjacent motif (PAM), whereas *As*Cas12a recognizes the T-rich, TTTV, and second, catalysis by *Sp*Cas9 results in blunt double-stranded DNA at three nucleotides 5′ to the PAM, whereas *As*Cas12a results in a staggered cleavage at 23 nt (target strand; T) and 18 nt (nontarget strand; NT) 3′ to the PAM [19]. Comparisons of these two RNA-programmed nucleases in the context of the *ω*1 gene of *S. mansoni* may inform decisions for functional genomics manipulations aiming to enhance efficiency of homology-directed repair (HDR) targeting genes expressed by the developing schistosome egg, and indeed other developmental stages [1,20,21,22].

## 2. Results

### 2.1. The Omega-1 Multicopy Gene, Guide RNAs, and Single-Stranded DNA Donor Templates

The gene-editing efficiency of *Sp*Cas9 and *As*Cas12a was compared in assays targeting the *ω*1 gene, which encodes a T2 ribonuclease that is secreted by the mature egg of *S. mansoni* into the surrounding host tissues and circumoval granuloma [23]. Specifically, the four copies of *ω*1, well annotated in WormBase Parasite [24] (update of 16 September 2021) as hepatotoxic ribonuclease; Smp_334170.2, Smp_334170.1, Smp_334240, and Smp_333930, were targeted for design of guide RNAs (gRNA) and donor-repair templates. These four copies reside on chromosome 1 of *S. mansoni*, with reverse strand positions at 3,980,364 to 3,982,676, 3,982,108 to 3,984,675, 3,992,964 to 3,995,246, and 3,908,953 to 3,911,250 (PRJEA36577), respectively. These copies share >99% nucleotide sequence identity, are ~2.3 kb in length, are highly AT-rich (65%), and encode an enzyme of 127 (Smp_334170.2, Smp_334240, and Smp_333930) or 115 (Smp_334170.1) amino acid residues in length. Smp_333220, also located on chromosome 1 at 3,925,420–3,927,716, but lacking annotation on the *S. mansoni* annotated genome version 7, year 2021, was also included. Smp_333220 shares >99% chromosomal and mRNA sequence identity with the three *ω*1 copies (Appendix A). Guide RNAs for *Sp*Cas9 and *As*Cas12a designed with the assistance of the CHOPCHOP webtool; both target the same region of exon 1 of *ω*1, disrupting its catalytic active site histidine [25], with the programmed cleavage sites for both Cas enzymes in close proximity, separated by three nucleotides (Figure 1a,b).

We designed three single-stranded oligodeoxynucleotide (ssODN) donors for homology-directed repair (HDR) of the *ω*1 gene resulting from the programmed mutation. Given the differing PAM requirements of *Sp*Cas9 and *As*Cas12a, as well as the divergent structures of the DSB—blunt-ended by *Sp*Cas9 and sticky-ended by *As*Cas12a, the designs for the ssODNs differed for use with each nuclease. Based on earlier findings [1], the design of the ssODN donors included 3′ and 5′ homology arms (HA), each of 50 nt in length, complementary to the target strand for *Sp*Cas9 and for *As*Cas12a, for either the target strand, termed Cas9-ssODN, or for the nontarget-strand donor, termed Cas12a-NT-ssODN. Note that sequences identical to the PAM for *As*Cas12a and ~18 nt of the protospacer region were retained within the 5′-HA of Cas12a-T-ssODN (Figure 1b,c).

### 2.2. Absence of Trans Activity of Activated AsCas12a against the ssODN Donors

Previous reports dealing with *Lachnospiraceae bacterium* ND2006 Cas12a and *Francisella novicida* Cas12a describe indiscriminate *trans*-activity unleashed on ssDNA after gRNA activation and target binding of Cas12a [15,26]. Our goal here was to compare *Sp*Cas9 and *As*Cas12a in schistosomes and, given that we used an orthologous *As*Cas12a from *Acidaminococcus* sp. BV3L6 (IDT), in both KO and KI assays, we sought to confirm whether the *ω*1 gene-activated *As*Cas12a might also exhibit *trans*-activity against the donor single-stranded templates that we planned to use in KI assays, Cas12a-T-ssODN and Cas12a-NT-ssODN, in addition to RNA-guided double-stranded DNA cleavage activity.

Accordingly, we carried out in vitro *As*Cas12a nuclease activity assays using RNPs targeting a 426 bp amplicon of the exon 1 of *ω*1 (Figure 2a–c). These assays confirmed that the crRNA was highly active and following incubation at 37 °C for 30 min had cleaved the double-stranded amplicon at the programmed cleavage site into two fragments of 265 and 161 bp, as expected (Figure 2b; Agilent Bioanalyzer gel). In contrast, the activated *As*Cas12a-RNP complex failed to digest either of the single-stranded oligodeoxynucleotide (ssODN) templates, ssODNs Cas12a-T-ssODN or Cas12a-NT-ssODN, after 120 min at 37 °C (Figure 2c). As seen in the gel image from BioAnalyzer, the ssODNs remained intact at 124 nt. Single-stranded DNA standards ranging in size from <40 to 200 nt were included in the assay (Figure 2c). In summary, the findings suggested that omega-1 target gene-activated *As*Cas12a would not exhibit indiscriminate *trans*-nuclease activity against the ssODNs within schistosome cells (although an in vitro surrogate assay cannot faithfully mimic conditions within a viable schistosome egg).

### 2.3. Quantification of Transgene Copy Number by External Standard Curve

An external standard curve design was employed to estimate transgene copy number in genomics DNA of the schistosome eggs (Figure 2e,f). The standard curve is based on simple linear regression (y = −4.0964ln [10] + 44.821, R^2^ = 0.9923) established with logarithm-10-transformed initial DNA input, plasmid pCR4, which encodes the six-stop codon cassette [1], as the dependent and the Ct value from the qPCRs as the independent variable (not shown). We estimated the transgene copy number on the 5′ and 3′ of KI site in *ω*1 by converting the measured Ct values to log copy numbers using the equations obtained from the standard curves. Subsequently, dividing the means by the antilog of each result, we arrived at an estimate of the transgene copy number. A 790 bp product amplified for the control wildtype sequence and amplicons of 235 bp and 182 bp for the 5′ and 3′ flanking regions, respectively, of the KI genomic DNA containing the six-stop codon transgene were analyzed, which revealed 3′-side transgene integration values of 3.61 ± 0.25, 4.44 ± 0.84, and 4.19 ± 1.22 from the KI_Cas9, KI_Cas12a-NT-ssODN, and KI_Cas12a-T-ssODN treatment groups, respectively (*p* = 0.4057, = 0.6264, and = 0.9155; NS; one-way ANOVA, multiple comparisons, 95% CI for differences of the means) (Figure 2e). However, the presence of asymmetrical HDR was revealed by significant differences among the copy number values for the 5′ side transgene integration estimates: 2.72 ± 0.42, 5.06 ± 0.54, and 1.64 ± 0.39 for the KI_Cas9, KI_Cas12a-NT-ssODN, and KI_Cas12a-T-ssODN groups, respectively (Figure 2f; *p* = 0.0214, *p* = 0.0001 and *p* ≤ 0.0001, *, ***, ****, respectively; one-way ANOVA, multiple comparisons).

### 2.4. Estimation of CRISPR Efficiency of SpCas9 and AsCas12a

RNP complexes of *Sp*Cas9-sgRNA and *As*Cas12a-CRISPR RNA (crRNA) targeting *ω*1 were delivered to the schistosome eggs by electroporation [1]. The efficiency of programmed mutation was assessed by TIDE analysis of Sanger sequence chromatograms [26,27], followed by targeted amplicon NGS and analysis of the reads using CRISPResso2 [28,29]. Twelve independent biological replicates were carried out by TIDE analysis. *As*Cas12a-programmed KO led to significantly more insertion–deletion (indels) than were induced with *Sp*Cas9 (*p* < 0.001, paired, two-tailed *t* = 7.450, df = 11) (Figure 2g). The mean CRISPR efficiency for KO with *Sp*Cas9 was 0.89% ± 0.43 (range, 0.30–1.31%) and 1.83% ± 0.29 (range, 1.39–2.54%) with *As*Cas12a, when compared with the control (mock) groups. Programmed mutation by *Sp*Cas9 was more efficient in the presence of donor ssODN, i.e., KI versus KO groups, 1.43 ± 0.4, range 0.51–2.31% (Figure 2g,h). By contrast, differences in mutation efficiency were not apparent between the KO and KI groups using *As*Cas12a; 1.75% ± 0.63 (range 1.11–2.69%) and 1.54% ± 0.49 (range 1.82–2.20%) for Cas12a-NT ssODN and Cas12a-T ssODN, respectively. From our result, there was no statistical significance of HDR efficiency between Cas12a-NT ssODN and Cas12a-T-ssODN in CRISPR/Cas12a KI experiment (Figure 2h). For deeper and high throughput analysis, both on target KO and KI, efficiency was also investigated by CRISPResso2 from pooled-genomic DNA samples by target amplicon reads by MiSeq 2 × 250 bp configuration sequencing (Illumina).

### 2.5. Estimates of Indels from NHEJ and/or HDR Using NGS Reads from Targeted Amplicons

Pooled gDNAs (above) were also used as templates for targeted amplicons for NGS and CRISPResso2 employed to assess the efficiency of DNA repairs; nonhomology end-joining (NHEJ) and transgene KI from homology-directed repair (HDR) from the NGS reads. Comparison of the mutations among all four copies of the *ω*1 gene also were undertaken. Whereas TIDE analysis using Sanger direct sequencing chromatograms was used above to estimate % efficiency of CRISPR manipulations in individual biological replicates, here, targeted amplicons from pools of gDNAs from the biological replicates were used to acquire libraries of reads (Amplicon EZ, Genewiz). More than 150,000 reads from each NGS library were obtained. Analysis of the reads using CRISPResso2 provided increased sensitivity of detection of programmed gene editing than TIDE, as expected [30]. Specifically, there were 12.26%, 1.66%, and 0.12% modified sequences (NHEJ) for KO-Cas9, and 9.27%, 1.33%, and 0.28% of NHEJ for KO-Cas12a, when compared with the WormBase Parasite reference sequences (identical amplicon from Smp_334170.1, Smp_334170.2, Smp_334240, Smp_333930, and Smp_333870, respectively (Figure 3a,b). A small number of reads were assessed to be ambiguous by CRISPResso2, 0.49% and 0.93% for KO-Cas9 and KO-Cas12a, respectively (Figure 3a,b). Deletions were the only type of indel detected, for both *Sp*Cas9 and *As*Cas12a. In similar fashion to other studies [31,32,33], deletion of large sizes resulted from programmed *As*Cas12a editing (from 2 to 26 bp deletion size) at the programmed DSB, compared to 3 nt at the DSB programmed from *Sp*Cas9 (Figure 3c,d provide representative allele sequences). Both enzymes showed a preference to mutate the reference Smp_334170 copy, and also Smp_334240, rather than the Smp_333930 and/or Smp_333870 copies of omega-1. The numbers of indels arising from NHEJ with *Sp*Cas9 and with *As*Cas12a both increased in the presence of an ssODN donor repair template. The total NHEJ percentages were 15.67%, 28.71%, and 21.43% from KI-Cas9-NT-ssODN, KI-Cas12a-NT-ssODN, and KI-Cas12a-T-ssODN, respectively (Figure 4a–c). Deletion size as determined by CRISPResso2 for the KI-Cas9 group was larger than for KO-Cas9, increasing up to 13 bp length deletions. Similarly, base deletion size from KI-Cas12a was larger than KO-Cas12a, with deletions of 42 bp in size detected (Figure 4d–f). Levels of ambiguous results were estimated at 1.66%, 1.03%, and 0.73% from KI-Cas9, KI-Cas12a-NT-ssODN, and KI-Cas12a-T-ssODN, respectively.

To estimate the frequency of HDR events, we included the donor sequence as a parameter in the CRISPResso2 analysis. More HDR events mediated by CRISPR/*Sp*Cas9 were detected than with CRISPR/*As*Cas12a; 17.07% vs. 14.56% and 12.37%, in similar fashion to our earlier findings [34]. However, in *Sp*Cas9 samples, more imperfect HDR events were seen (0.94%) than in the *As*Cas12a groups (0.41% and 0.68%) (Figure 4a–h). Frequently, imperfect HDR event alleles from both Cas nucleases were missing several residues (2–5 nt) on the 5′ side of the programmed DSB. We note that this mutation repair outcome may have impacted quantification of the 5′ KI transgene copy number, especially in KI-Cas12a-T-ssODN samples which contained 5 nt imperfect HDR repair on the six-stop codon cassette (primer probing sequence). Imperfect HDR at the 5′ side might result from synthesis-dependent strand annealing [35].

### 2.6. Transcription of ω1 Interrupted by Programmed Mutation by SpCas9 and AsCas12a

Primers specific for the four annotated copies of *ω*1 gene (Appendix A) were employed to investigate transcription by real-time quantitative PCR. Relative expression levels of *ω*1 mRNA in negative controls and experimental KO or KI eggs were compared with wild type. Significantly reduced levels of *ω*1 were seen in eggs following the experimental CRISPR manipulations for gene KO and KI. Transcription of *ω*1 was reduced, as follows: 63.76% ± 5.57, 76.88% ± 11.33, 71.81% ± 10.91, 98.69% ± 0.44, and 80.04% ± 0.09 in the KO_SpCas9, KI_SpCas9, KO_Cas12a, KI_Cas12a-NT-ssODN, and Cas12a-T-ssODN groups, respectively. Notably, LE subjected to KI_Cas12a_NT-ssODN retained <2% *ω*1 transcript levels of the mock control (Figure 5a), and the level in the KI_Cas12a-NT-ssODN group was significantly lower than the KI_Cas12a-T-ssODN group, KI_Cas9-ssODN. Significant differences were not apparent between the KO_Cas9 and KO_Cas12a groups, generally. There were no statistic differences in levels of transcription among the negative control groups (Figure 5a). Differences among treatment groups were assessed using ANOVA with multiple comparisons (*p* ≤ 0.01 to ≤0.0001; *n* = 4; *F* = 11.77).

### 2.7. Protein Levels Ascertained by Western Blotting

An anti-recombinant *ω*1 rabbit IgG was used to investigate levels of omega-1 in lysates (soluble egg antigen; SEA) of LE. SEA (200 ng) from the three controls, i.e., the mock, enzyme-only, and mixed gRNAs-only groups, and from the KO and KI groups with *Sp*Cas9 or *As*Cas12a was size-fractionated by SDS-PAGE and transferred to PVDF and probed with the rabbit antibody (Figure 5b). Following densitometric analysis of the resulting signals, the levels of omega-1 remaining in the SEA were estimated at 64% ± 11.65, 43.88% ± 6.02, 49.27% ± 4.19, 37.9% ± 2.28, and 52.16% ± 10 in the KO_Cas9, KI_Cas9, KO_Cas12a, KI_Cas12a-NT-ssODN, and KI_Cas12a-T-ssODN groups, respectively. The differential intensity of *ω*1 expression was not diminished among the gRNA-only groups (Figure 5b,c). By contrast, *ω*1 protein levels were markedly diminished among all the experimental treatment groups when compared with the controls (*p* ≤ 0.0001) (ANOVA, *F* = 27.05, *n* = 4). Significant differences among for levels of *ω*1 among the KO and KI SEA samples were not apparent (Figure 5c).

## 3. Discussion

The omega-1 (*ω*1) extracellular T2 ribonuclease secreted by the mature egg of *S. mansoni* is a key modulator of pathogenesis and transmission of schistosomiasis [23,36,37,38]. Given the central role of the egg in disease and in the transmission [38] and given the potential of CRISPR/Cas gene editing to advance understanding of this and other neglected tropical diseases, access to omega-1-negative eggs can provide information to confirm the role of this ribonuclease in disease transmission. Functional knockout (KO) of the *ω*1 gene and the resulting immunologically impaired phenotype have showcased the novel application of CRISPR/*Sp*Cas9 and its utility for functional genomics in schistosomes [4,22,39,40]. In view of this progress, and the demonstrated tractability of *Sp*Cas9-catalyzed KO of the omega-1 gene, we here compared the performance of a second RNA-guided Cas nuclease, the *As*Cas12a nuclease of *Acidaminococcus* sp., in terms of efficiency and precision with those for *Sp*Cas9.

Guide RNAs for both *Sp*Cas9 and *As*Cas12a were designed to anneal to protospacer regions that are essentially at the same target site; the programmed DSBs were separated by only three nucleotide residues, on the nontarget strand of the target gene. The optimal protospacer-adjacent motif for *As*Cas12a, TTTV [11,41], is expected to occur abundantly in the AT-rich genome of *S. mansoni* [42]. Programmed DSB catalyzed by *As*Cas12a results in a staggered strand break of the target gene sequence, which can enhance efficiency of HDR [41]. For RNPs assembled from the nuclease and cognate crRNA, *As*Cas12a was significantly more efficient for gene knockout than *Sp*Cas9, as assessed by TIDE, presumably reflecting NHEJ-catalyzed repair and gene silencing.

We also investigated HDR-mediated gene editing with *Sp*Cas9 and *As*Cas12a delivered by electroporation in the presence of the RNPs and donor repair templates. Each donor ssODN template for *Sp*Cas9- and *As*Cas12a-programmed KI included a transgene cassette of 24 nt in length encoding six-stop codons [1] but in which the homology arms differed slightly on the nontarget strand due to the discrete positions of the predicted DSB for each nuclease. For programmed KI catalyzed by CRISPR/*As*Cas12a, we also included, as the donor template, the reverse complement sequence of the six-stop codons cassette flanked by the target stand sequence as the homology arms. This additional donor was included in our analysis, given concern with respect to the known indiscriminate trans-activity by activated *As*Cas12a against single-stranded DNA [15], and especially for the ssODN with NT homology arms which retains the PAM and partial (~18 nt) protospacer sequence [41,43,44]

RNA-guided *Sp*Cas9 cleavage led to higher levels (~4% higher) of precise knockout compared to that obtained with *As*Cas12a but markedly less precise knock-in (~6–13%). Deletions were the frequent mutation type seen with both *Sp*Cas9 (deletions of 3 nt in length) and *As*Cas12a (deletions of 2 to 26 nt in length) due to NHEJ repair. Using conditions optimized previously for *Sp*Cas9 KI [1], CRISPR/*Sp*Cas9 induced higher (~3%) HDR levels than CRISPR/*As*Cas12a to insert the transgene for both Cas12a-NT-ssODN and Cas12a-T-ssODN donors. Both *Sp*Cas9-and *As*Cas12a-edited eggs exhibited reduced levels of transcription and expression of omega-1. A spectrum of chromosomal alleles was apparent following programmed knock-in of the ssODN repair templates. Both target strand and nontarget strand donors were provided. Whereas accurate HDR was catalyzed by both *Sp*Cas9 and *As*Cas12a, precise repair accounted for only a minority of the payload integrations. In some cases, the two sides of the DSB programmed by *As*Cas12a were repaired by different pathways, e.g., one side repaired by HDR and the other by NHEJ, since mutant alleles that included only the left or only the right side of the transgene were observed [45]. Moreover, accurate HDR was significantly more frequently detected in the presence of the nontarget strand compared with target strand donor.

These findings may have been expected based on reports with other cells and species that indicate that integration of a repair template is not always precise [46,47]. In addition, *Sp*Cas9 may lead to higher levels of HDR correction than *As*Cas12a in some contexts, including in HEK293T and pluripotent stem cells where misintegration patterns indicate that the two sides of a DSB are repaired by separate cellular repair pathways including the involvement of microhomology as a driver of the misintegrations [34].

In addition to its guide RNA-programmed gene editing and DSB activity, *As*Cas12a and related Type 2 subtypes V and VI nucleases display target substrate bound, nonspecific *trans*-activity against ssDNA, dsDNA, and/or RNA substrates [48,49]. Accordingly, we speculated that activated *As*Cas12a associated with the target exon 1 site in omega-1 may digest the donor ssODN provided as the repair template. If so, our investigation of comparative efficiency of *As*Cas12a vs. *Sp*Cas9 catalyzed knock-in would have been compromised. To address this possibility, we performed a series of assays in vitro with both nonactivated and activated *As*Cas12a. Under the conditions and time frame of the reactions, commercially-sourced *As*Cas12a, 37 °C, 500 mM NaCl, 20 mM sodium acetate, 0.1mM EDTA, 0.1 mM Tris-phosphine solution, and 120 min reaction duration, we failed to detect degradation of the ssODNs. Although our assays do not faithfully reproduce the physiological environment of the transfected schistosome eggs in vivo, we failed to detect indiscriminate *trans*-activity against the ssODNs. However, lack of apparent *As*Cas12a *trans*-activity in vitro does not preclude that indiscriminate *trans*-activity does not take place within the nucleus of the cells within the egg. The RNP may not have been retained at the programmed target site, perhaps degraded by cellular enzymes and/or displaced by the cellular repair machinery following cleavage before *trans*-activity damaged the donor ssODN or other phenomena including protection of cellular DNA by histones. Nonetheless, both *Sp*Cas9 and *As*Cas12a delivered similar knock-in performance in these schistosome eggs.

Whereas enhanced mutation efficacy of *As*Cas12a over *Sp*Cas9 has been reported, the reverse outcome also has been seen [50,51]. Comparisons among studies are not straightforward because of differences in species, target genes, delivery method, and temperature, among other confounders. In addition, three orthologues of *As*Cas12a, *Fn*Cas12a from *Francisella novicida* U112, *Lb*Cas12a from *Lachnospiraceae bacterium* ND2006, and *As*Cas12a from *Acidaminococcus* sp. BV3L6 have been widely studied in diverse eukaryotes [8] and other orthologues also have been reported [52,53]. *As*Cas12a is active at 37 °C whereas some others are catalytically active at lower temperatures, which has enabled studies with zebrafish [54], insects, and maize [53], and environmental and diagnostic biosensing [15,55,56].

*As*Cas12a provided enhanced programmed knockout activity compared to *Sp*Cas9 at a specific target site in the omega-1 gene. By contrast, both nucleases delivered similar performance for HDR-based transgene insertion. Although these findings do not enable extrapolation to performance of *As*Cas12a for CRISPR/Cas-based gene editing of other schistosome genes, they confirm that Cas12a is active in schistosomes and expand the available nuclease tools for functional genomics in helminth parasites [22,57]. Optimization of the *As*Cas12a-based approach with respect to ratios of guide RNA:*As*Cas12a and/or ssODN, format of donor(s), and/or temperature will be informative [15,44,58]. We anticipate that *As*Cas12a will find increasing use in functional genomes for AT-rich genomes, including schistosomes and other platyhelminths [59]. As transgene insertion into the zygote within the newly released egg has already been described and is shown to enable establishment of lines of transgenic schistosomes, advances with gene editing approaches to this key developmental stage of the schistosome life cycle portend progress with the derivation of informative gain-of-function lines of transgenic schistosomes.

## 4. Materials and Methods

### 4.1. Ethics Statement

Mice experimentally infected with *S. mansoni*, obtained from Schistosomiasis Resource Center (SRC) at the Biomedical Research Institute (BRI), MD were housed at the Animal Research Facility of the George Washington University (GW), which is accredited by the American Association for Accreditation of Laboratory Animal Care (AAALAC no. 000347) and has an Animal Welfare Assurance on file with the National Institutes of Health, Office of Laboratory Animal Welfare, OLAW assurance number A3205-01. All procedures employed were consistent with the Guide for the Care and Use of Laboratory Animals. The Institutional Animal Care and Use Committee (IACUC) at GW approved the protocol used for maintenance of mice and recovery of schistosomes. Swiss albino mice were euthanized seven weeks after infection with *S. mansoni*, livers were removed at necropsy, and schistosome eggs recovered from the livers [60]. The liver eggs, termed “LE”, population is expected to include eggs of a spectrum of age ranging from newly released by the female schistosome through to mature eggs containing a fully developed miracidium [61]. LE were maintained in DMEM medium supplemented with 20% heat-inactivated fetal bovine serum (FBS), 2% streptomycin/penicillin at 37 °C under 5% CO_2_ in air for 18–24 h before use in gene editing assays (below) [1,60,62,63,64].

### 4.2. CRISPR/Cas Target Design and Single-Stranded DNA Donors

The *ω*1-gene-specific guide RNA (gRNA) for *Sp*Cas9 and *As*Cas12a were designed to target all four copies of the hepatotoxic ribonuclease *ω*1 gene. All four are located in the *S. mansoni* genome updated annotation of version 7 as the data accessed on 3 October 2021 (updated 16 September 2021 on Worm Base ParaSite, https://parasite.wormbase.org/Multi/Search/Results?species=all;idx=;q=Hepatotoxic%20ribonuclease%20omega-1;site=ensemblunit&filter_species=Schistosoma_mansoni_prjea36577); Smp_334170.1, Smp_334170.2, Smp_334240, and Smp_333930 by an online CRISPR design tool, CHOCHOP [65,66,67]. There are 99–100% sequence and coding sequence identity among those copies (Appendix A), which encode an enzyme of 115 amino acids in length, respectively. These *ω*1 copies are located on chromosome 1 at nucleotide (nt) positions 3,980,364 to 3,984,675, 3,992,964 to 3,995,246, and 3,908,953 to 3,911,250, respectively. The guide RNA (gRNA) targets exon 1 of *ω*1 complimentary to the catalytic active site histidine (within FRKHEFEKHGLCAVEDPQV) codons for the gRNAs programming both *Sp*Cas9 and *As*Cas12a. In addition, the gRNAs exhibited similarly high predicted CRISPR efficiency scores along with an absence of predicted off-target cleavages, as predicted using the CHOPCHOP algorithm using Smp_334170.2 as template (Figure 1a,b). Specifically, the sequence of the gRNA used with *Sp*Cas9 was 5′-GCATGGTTTGTGTGCAGTTG-3′, 20 nt in length, and that for *As*Cas12a was 5′-AAAAGCATGGTTTGTGTGCA-3′, 20 nt on exon 1 of Smp_334170.2, Smp_334240, and Smp_333930 or exon 4 of Smp_334170.1 (Appendix A). The expected cut sites of two nucleases are discrete, being 3 and 2 nt apart from each other on the nontarget (NT) and target (T) strands, respectively (Figure 1a).

Single-stranded oligodeoxynucleotide donors (ssODN) were designed to insert the 24 bp encoding six-stop codons into the programmed cleavage target. The homology arms used in the ssODN donor with *Sp*Cas9 was the nontarget strand flanking both the 3′ and 5′ sides of the blunt-ended DSB, termed here Cas9-ssODN, as described previously [1]. The homology arms of the two donor ssODNs used with *As*Cas12a (cut at 18 nt and 23 nt on NT and T strands, leading to a sticky-ended DSB) were either the target stand or the nontarget strand sequences (Figure 1b,c), here termed Cas12a-NT-ssODN (HA complementary to T strand at 23 nt cut site) and Cas12a-T-ssODN (containing HA reverse complementary reverse to NT strand at 18 nt cut site), respectively. The complementary reverse of the 24 nt stop codon cassette was also used in the Cas12a-T-ssODN donor, aiming for HDR insertion of the stop codons in the translated forward strand of the gene (Figure 1c).

### 4.3. CRISPR Reagents

Alt-R^®^ CRISPR-*Sp*Cas9 and Alt-R^®^ CRISPR-*As*Cas12a reagents, including Alt-R^®^ S.p. HiFi Cas9 nuclease V3, synthetic gRNA (sgRNA) [9,65,66], Alt-R^®^ A.s. Cas12a nuclease V3 and crRNA, ssODN synthesis (Ultramer DNA oligos), and electroporation enhancer buffers were from Integrated DNA Technologies (IDT) (Coralville, IA, USA). Figure 1c provides the nucleotide sequences of the gRNAs and donor ssODNs.

### 4.4. Ribonucleoprotein (RNP) Assembly and Delivery by Electroporation

Stock solutions of the gRNAs for use with *Sp*Cas9 and *As*Cas12a and of the ssODNs were prepared at 1.0 µg/µL in Opti-MEM medium (Sigma, St. Louis, MO, USA). Each gRNA was combined with the respective nuclease at 1:1 ratio to form an RNP complex as described [1]. The mixture was incubated at room temperature for 15 min, after which 3 µL of Alt-R^®^
*Sp*Cas9 or *As*Cas12a electroporation enhancer was added. The mixture was transferred to a chilled 4 mm electroporation cuvette (BTX), containing ~5000 LE in 100 µL Opti-MEM. LE in the presence of RNPs and the other reagents were subjected to square wave electroporation with a single pulse of 125 volts for 20 ms (ElectroSquarePorator, ECM830, BTX San Diego, CA, USA). In KI assays, 6 µg of ssODN was included in the cuvette before electroporation. Following delivery of the RNPs, LE were maintained for 10 days at 37 °C, 5% CO_2_ in air. In addition to experimental, gene-edited groups, control groups included LE electroporated in Opti-MEM (mock), mixed-Cas enzymes only, mixed-sgRNAs only, and mixed-ssODNs only. Experimental and control groups were otherwise treated similarly.

### 4.5. PCR to Detect a Multiple Stop Codon Containing Transgene

Genomic DNAs were extracted from LE using the DNAzol^®^ RT reagent (Molecular Research Center Inc., Cincinnati, OH, USA), and the concentration and purity determined by spectrophotometry (Nanodrop 1000, Thermo Fisher Scientific Inc., Waltham, MA, USA). The HDR target site(s) on either the 5′ or 3′ KI transgene were amplified independently with two pairs of primer (Figure 2a): (1) 5′KI: 5′int-F and 24 nt-6stp-R primers and (2) 3′KI: 24 nt-6stp-F and 3′int-R using the GoTaq DNA polymerase mix (Promega, Madison, WI, USA) and 200 nM of each primer. The thermal cycling included denaturation at 95 °C, 3 min, followed by 30 cycles of 94 °C, 30 s, 56 °C, 30 s and 72 °C, 30 s, and a final extension for 5 min at 72 °C. Thereafter, the amplicons (10 μL) were size-separated by agarose gel electrophoresis, stained with ethidium bromide, and imaged using the ChemiDoc Imaging System (Bio-Rad, Hercules, CA, USA). The expected sizes for the 5′KI and 3′KI amplicons were 235 and 182 bp, respectively (Figure 2d). A positive-control PCR to amplify ~790 bp flanking the DSB was included.

### 4.6. Real-Time qPCR to Estimate Transgene Copy Numbers

An SYBR green quantitative PCR-based approach was used to estimate transgene copy numbers using two pairs of primers: 5′KI and 3′KI, as detailed (Figure 2a,d), with the SsoAdvanced Universal SYBR Green Supermix (Bio-Rad). Briefly, one ng of genomic DNA (gDNA) was amplified with 10 µL of 2 × SsoAdvanced Universal SYBR Green Supermix and 200 nM of each primer, using the CFX Connect Real-Time PCR Detection System (Bio-Rad). PCR conditions included denaturation at 95 °C, 3 min, followed by 30 cycles of 94 °C, 30 s, 56 °C, 30 s and 72 °C, 30 s and a final extension at 72 °C for 5 min, with the signals collected at the annealing step in each cycle. Analysis of transgene copy number in the pooled gDNAs from LE involved comparison of the qPCR signals obtained using a 10-fold serial dilution series of known quantities of the 24 nt six-stop codon transgene in 235 bp (5′ integration PCR fragment) ligated-pCR4-TOPO plasmid DNA (size 4191 bp) to establish a standard curve [68]. The standard curve was based on a linear regression generated using logarithm-10-transformed initial DNA input as the dependent variable and the Ct number of the qPCR signal as the independent variable (Ct values in each dilution were measured in triplicate). Ct numbers were used to fit into the linear regression to derive the estimation of transformed DNA amount for the transgene [69,70]. The concentration of the 5′KI-transgene-ligated-pCR4-TOPO plasmid was measured (NanoDrop 1000), and the corresponding copy number was calculated as follows:(1)$$\mathrm{DNA}\;\mathrm{copy}\;\mathrm{number}=\frac{6.02\times 10^{23}\;\left(\mathrm{copy}/\mathrm{mol}\right)\times \;\mathrm{DNA}\;\mathrm{quantity}\;\left(g\right)}{\mathrm{DNA}\;\mathrm{length}\;\left(\mathrm{bp}\right)\times 660\;\left(g/\mathrm{mol}/\mathrm{bp}\right)}$$

Data were collected from four biological replicates. The 5′KI and 3′KI transgene copy numbers from the *Sp*Cas9 and *As*Cas12a treatments were plotted using GraphPad Prism v9 (Figure 2e,f).

### 4.7. Tracking of INDELs by Decomposition (TIDE) Analysis

Genomic DNAs from LE exposed to the RNPs were isolated as described [1]. The target omega-1 locus in the replicates of gDNAs was amplified using a control primer pair (Figure 2a). The amplicon of 790 bp in size from control and experimental groups, both KO and KI focused manipulations, was purified by NucleoSpin^®^ Gel and PCR clean-up (Macherey-Nagel, Bethlehem, PA, USA) after which the nucleotide sequence was determined by the Sanger method (Genewiz, South Plainland, NJ, USA). Sanger sequence traces from the twelve independent replicates of the experimental groups for both *Sp*Cas9- or *As*Cas12a- [27]-catalyzed gene editing manipulations were analyzed and compared with reads from the control (mock) group using the tracking of indels by decomposition (TIDE) algorithm [71,72,73]. TIDE provides a rapid and informative assay and also informs decisions on whether proceed to more detailed high-throughput, next-generation sequencing (NGS) of targeting amplicons [74].

### 4.8. On-Target Amplicon Next-Generation Sequencing

Subsequently, the Amplicon-EZ next-generation sequencing approach (Genewiz) was used for deeper coverage of the *ω*1 coding sequence mutations and provided >150,000 reads per sample. Amplicons of 426 bp obtained with the primers bearing Illumina partial adapter sequences at the 5′ end-NGS-F and NGS-R (Appendix A) flanking the programmed DSBs for CRISPR/*Sp*Cas9 and CRISPR/*As*Cas12a were sequenced using Illumina chemistry. Partial Illumina adapters were added to 5′ end of the forward and reverse primers forward sequencing read: 5′-ACACTCTTTCCCTACACGACGCTCTTCCGATCT-3′, reverse sequencing read: 5′-GACTGGAGTTCAGACGTGTGCTCTTCCGATCT-3′. The amplicons were purified using the NucleoSpin^®^ Gel and PCR clean-up kits from Macherey-Nagel (Bethlehem, PA, USA), after which quantification and qualification of purified band(s) were undertaken using the Agilent DNA 1000 kit and BioAnalyzer (Agilent, Santa Clara, CA, USA). Illumina NGS was performed with 2 × 250 bp configuration sequencing without fragmenting the amplicons. Raw reads in FASTQ format were analyzed for mutations resulting from NHEJ and HDR by CRISPREsso2 [29] using the reference *ω*1 sequences, Smp_334170 and Smp_333930, including Smp_333870 (this latter is annotated as an uncharacterized protein but shares >99% identity) (Appendix A). The default parameters were used to estimate indel and HDR events. The sequence reads from the Amplicon-EZ runs are available at GenBank Bioproject PRJNA415471, BioSample SAMN07823308, SRA study SRP126685, accessions SRR: 13498338-13498342, 15971309-15971320, and 6374209-6374210.

### 4.9. Quantitative Real-Time PCR

Total RNA of LE was isolated using the RNAzol^®^ RT reagent (Molecular Research Center) according to the manufacturer’s instruction. RNA concentration and purity were determined (Nanodrop 1000 spectrophotometer, Thermo Fisher). Ten nanograms of RNA was treated with DNase I to remove residual DNA then converted to cDNA by Maxima First Strand cDNA Synthesis Kit (Thermo Fisher Scientific). The *ω*1-specific product was amplified using SsoAdvanced Universal SYBR Green Supermix (Bio-Rad) on CFX96 real-time PCR (Bio-Rad) using the primers as described [1], *ω*1-RT-F and *ω*1-RT-R (Appendix A). These primers amplify all annotated *ω*1 transcript copies; Smp_334170.1, Smp_334170.2, Smp_334240, and Smp_333930 (Appendix A). The PCR reaction was denatured at 95 °C for 30 s, 40 amplification cycles each consisting of denaturation at 95 °C for 15 s and annealing/extension at 60 °C for 30 s. The output was analyzed in CFX manager software (Bio-Rad). Relative expression of *ω*1 was calculated using the 2^−∆∆Ct^ method and normalized to schistosome GAPDH; *Sm*GAPDH (Smp_0569701) expression using specific primers as in Appendix A [75]. Data of four biological replicates were compared to wild type LE and fold change reported as mean ± SE (*n* = 4).

### 4.10. Western Blot Analysis

Soluble egg antigen (SEA) of LE was extracted by mechanical lysis (Funakoshi motorized pestle; Diagnocine, Hackensack, NJ, USA) in RIPA buffer (Millipore Sigma) supplemented with protease inhibitors (Protease Inhibitor Cocktail I, Millipore Sigma), as described [1]. The lysate was clarified by centrifugation at 13,000 rpm, 15 min, 4 °C after which protein concentration was determined by the Bradford method [76]. Ten micrograms of SEA were size-separated through SDS-PAGE polyacrylamide gradient gels, 4–12% gel, *w*/*v* (Bolt Bis-Tris Plus, Invitrogen, Thermo Fisher Scientific) before transferring to PVDP membranes (Transblot Turbo Transfer System, Bio-Rad). Polyclonal anti-*ω*1 IgG antibodies were purified from the rabbit anti-recombinant *ω*1 (r*ω*1) sera, raised as follows. Two NZ white rabbits, E12983 and E12984, were immunized with 30 μg of recombinant omega-1 RNase T2 of *S. mansoni* (r*ω*1) (produced in planta by agroinfiltration of leaves of *Nicotiana benthamiana* [77]) in Complete Freund’s Adjuvant and boosted four times at two-week intervals using 15 μg r*ω*1 in Incomplete Freund’s Adjuvant each time. Blood was collected 66 days after the first immunization and IgGs were isolated from sera using an immobilized protein A ligand. The concentrations of the purified antibodies were 1.31 mg/mL (E12983) and 1.51 mg/mL (E12984). These antibodies have been assigned Research Resource Identifier (RRID) AB_2889886 [78]. PVDF membranes were probed with the E12983 IgG antibody, diluted 1:1000, for 18 h at 4 °C in PBS-Tween (0.5%), 5% nonfat powdered milk, washed 3 times in PBS-Tween (wash buffer), and probed with goat anti-rabbit IgG (H + L) horse radish peroxidase conjugate (Life Technologies, Thermo Fisher Sciences, San Diego, CA, USA), diluted 1:5000, for 60 min at 23 °C with gentle shaking. Following washing (3 times in wash buffer), the membrane was exposed to a peroxidase substrate (Clarity Western ECL Substrate, Bio-Rad) for 5 min at 23 °C, after which chemiluminescence was quantified and imaged (Chemi-Doc MP System, Bio-Rad). The Image Lab software (Bio-Rad) was used to quantify the *ω*1 signals. The percentage of *ω*1 expression from the experimental and control groups was established by comparison with that of wild type SEA, assigned as 100%.

### 4.11. Single-Stranded DNA Donors as a Potential Substrate for Indiscriminate Trans Activity of Activated AsCas12a

Previous reports reveal that the *trans*-activity of wild type *As*Cas12a prefers ssDNAs over dsDNAs and was inactive toward dsDNA in the presence of magnesium (II) ions [79]. The *As*Cas12a exhibited not only endonuclease activity on both dsDNA, but also *cis*- and *trans*-activity [80]. Accordingly, to investigate whether activated *As*Cas12a could degrade the ssODN donors used in this study by indiscriminate *trans*-activity, several in vitro *As*Cas12a-RNP digestion assays were carried out. These included the dsDNAs containing the *As*Cas12a target sequence used in this study and 5′PAM (TTTG) as positive enzymatic activity. A 426 bp dsDNA template was amplified using the NGS study primers. Activity of *As*Cas12a-RNP against the dsDNA template released fragments of 265 and 161 bp (Figure 2b) at 37 °C for ≥30 min, as predicted in 1× NEB Buffer 2.1 (500 mM NaCl, 20 mM sodium acetate, 0.1 mM EDTA, 0.1 mM tris(2-carboxyethyl) phosphine (TCEP), 50% glycerol). *As*Cas12a activity was inactivated by the addition of Proteinase K (1 µL at 20 mg/mL) at 56 °C for 10 min. Electropherograms of the digestion products were obtained using the BioAnalyzer (Agilent DNA1000) to visualize the size of templates and fragments.

To investigate indiscriminate *trans*-cleavage of ssODNs, similar in vitro digestion reaction was carried out, as above, but with the addition of 50 ng of either *As*Cas12a-NT-ssODN or *As*Cas12a-T-ssODN. A mixture of *As*Cas12a-RNP complex in 1× NEB Buffer 2.1 and ssODN was incubated at 37 ℃ for 30, 60, and 120 min before terminating the reaction by addition of proteinase K at 56 ℃ for 10 min and visualizing as above. The ssDNA ladder (SimPlex Sciences, NJ, USA) ranging from 10–200 bp was included as the single-stranded DNA marker. *Trans-*activity of Alt-R *As*Cas12a V3-RNP complex was not apparent against the ssODNs under these conditions in vitro (Figure 2c).

### 4.12. Statistical Analysis

Statistical analysis was performed using GraphPad Prism version 9.1 (GraphPad Prism, La Jolla, CA, USA). Graphs were plotted as the mean with bars to show the standard error of the mean. The differences among groups were assessed by one-way ANOVA. *p* values of ≤0.05 were considered statistically significant.

### 4.13. Data Submission

Nucleotide sequences reads of the amplicons are available at Bioproject PRJNA415471 as recently access on 27 October 2021, https://www.ncbi.nlm.nih.gov/bioproject/PRJNA415471 and GenBank accession PRJNA415471, BioSample, SAMN07823308, SRA, SRP126685.

## Figures and Tables

**Figure 1 ijms-23-00631-f001:**
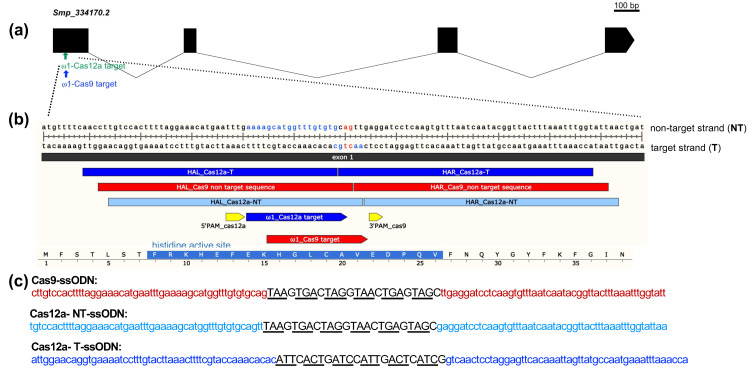
Schematic of the *S. mansoni* hepatotoxic ribonuclease *omega 1* (*ω*1) gene structure, target sites for the *Sp*Cas9 and *As*Cas12a, ssODN donors, and partial amino acid sequence with histidine active site: (**a**) diagram of the *ω*1 gene (Gene ID; Smp_334170.2 of *S. mansoni* annotated genome version 7_2021) and CRISPR/Cas target sites on coding sequence of the first exon coding for histidine active site, FRKHEFEKHGLCAVEDPQV. The relative positions of crRNA for CRISPR-*Sp*Cas9 (red arrow) and *As*Cas12a (blue arrow) are indicated, along with their 3′PAM (yellow arrow) and 5′PAM (yellow arrow), respectively; (**b**) programmed cleavage site for *Sp*Cas9/sgRNA ~3 nt upstream of PAM 3′TGG (red arrow); and cleavage site for *As*Cas12a/crRNA at 18–24 nt downstream of a prospective 5-’TTTV PAM (blue arrow). The programmed cleavage sites for both enzymes, *Sp*Cas9 and *As*Cas12a, were 3 to 6 nt distance from each other. Fifty residues upstream and downstream of the double-strand break (DSB) were included as the homology arms (HA) of the six-stop codon cassette transgene (payload/cargo). Because cleavage by *Sp*Cas9 liberates blunt-ended DNA, only a nontarget strand (NT) repair template was provided for HDR (red bar). Sticky-ended DNA results from cleavage by *As*Cas12a, and hence both nontarget and target strand (T) repair templates for HDR were provided (light and dark blue bars); (**c**) the DNA donor sequences of CRISPR/*Sp*Cas9 (red letters) and CRISPR/*As*Cas12a with nontarget (light blue letters) or target sequence (dark blue letters) HA where the Cas12a-NT-ssODN donor repairs the antisense DNA and Cas12a-T-ssODN donor repairs the sense stand.

**Figure 2 ijms-23-00631-f002:**
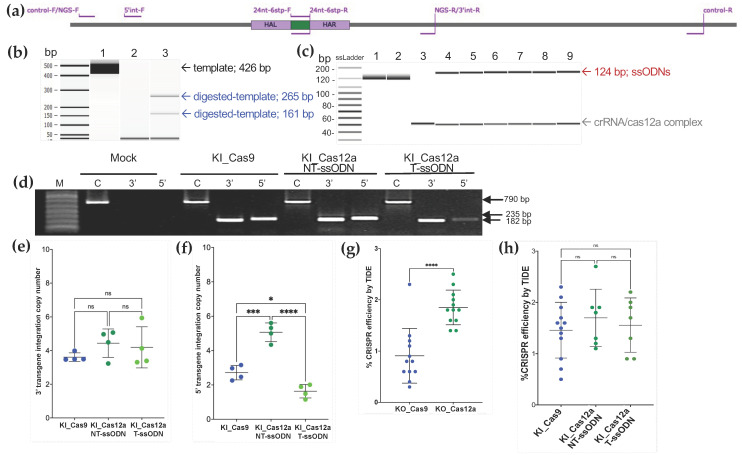
Knock-in by homology-directed repair: (**a**) The location of primers used for 5′- and 3′- transgene (purple lines) knock-in PCR (5′int-F with 24 nt-6stp-R and 24 nt-6stp-F with 3′int-R) and internal positive control (control-F and -R). The primers for next-generation sequencing are indicated as NGS-F and NGS-R. All primers are located outside of the homology arms; HAL and HAR (purple boxes) sequences to avoid false positive amplification from DNA donor. The 24 nt-6stp-F and -R primers are highly specific for the stop codon transgene and should not anneal to the wild type schistosome gene. The green box shows the position of the 24 nt six-stop codon transgene. (**b**) The *As*Cas12a target recognition activates specific 426 bp dsDNA amplicon template (lane 1) cleavage into 265 bp and 161 bp (lane 3) products. The crRNA/*As*Cas12a RNP complex of ~40 nt is seen in lanes 2 and 3; (**c**) No specific ssDNA cleavage was seen on the Cas12a-NT-ssODN or Cas12a-T-ssODN donors at 124 nt as in lanes 1 and 2, respectively. The crRNA/*As*Cas12a RNP (~40 nt in lanes 3–9) containing activated *As*Cas12a did not show indiscriminate ssDNA digestion at 30 min, 60, and 120 min in lanes 4–5, 6–7, and 8–9, respectively. Lanes 4, 6, and 8 show the uncut 124 nt of Cas12a-NT-ssODN (top band) with Cas12a-crRNA (lower band), and lanes 5, 7, and 9 show 124 nt of Cas12a-T-ssODN (top band) with Cas12a-crRNA (lower band); (**d**) The PCR detection of the integration sites in the transgenic eggs and control. The amplified control product (lanes C) for the wildtype sequence was 790 bp in length while those for the 5′ (lanes 5′) and 3′ (lanes 3′) flanking regions of the transgenic sequence were 235 bp and 182 bp due to KI of Cas-9 ssODN, Cas12a_NT-ssODN, and Cas12a_T-ssODN donor templates, respectively. Lane M is a 1kb Plus DNA ladder (Thermo Fisher Scientific, MA, USA); mock (only buffer electroporation) served as the negative control for transgene integration; (**e**,**f**) The estimation of transgene copy number in egg DNA were calculated by qPCR, comparing with known quantity of the 24 nt transgene in 235 bp (5′ integration PCR fragment) ligated-pCR4-TOPO plasmid DNA (size 4191 bp) standard curve; y = −4.0964ln(10) + 44.821, R^2^ = 0.9923 (data not shown). There was the confirmation of 3′ integration of 3.61 ± 0.25, 4.44 ± 0.84, and 4.19 ± 1.22 of transgene copy from KI_Cas9, KI_Cas12a_NT-ssODN, and KI_Cas12a_T-ssODN, respectively (one-way ANOVA, multiple comparisons, 95% CI of diff, *p*-value at 0.4057, 0.6264, and 0.9155, not significant). Asymmetric HDR was revealed by 5′ transgene integration of 2.72 ± 0.42, 5.06 ± 0.54, and 1.64 ± 0.39 copy number from KI_Cas9, KI_Cas12a_NT-ssODN, and KI_Cas12a_T-ssODN, respectively. The asterisk symbols *, ***, and **** indicate statistical significance at *p* = 0.0214, *p* = 0.0001, and *p* < 0.0001, respectively (one-way ANOVA, multiple comparison), while ‘ns’ indicates not statistically significant. The control primers used for amplification from both the control and experimental groups, followed by Sanger sequencing results, were monitored for efficiency of programmed CRISPR editing firstly by TIDE followed by NGS and CRISPresso2 of the NGS reads; (**g**,**h**) 5′int-F and 3′int-R are genome specific primers, and 24 nt-6stp-F and 24 nt-6stp-R are transgene-specific primers. From the findings, we confirmed that programmed CRISPR/*As*Cas12a gene editing was active in schistosome eggs, and higher than for *Sp*Cas9 by TIDE analysis (ANOVA, ****, *p* < 0.0001, *n* = 12). The estimation of CRISPR efficiency from the 12 biological replicates (each black dot in panels g and h) KI_Cas9 (wide range of CRISPR efficiency from 0.46–2.23 among 12 biological replicates) was higher than for KO-based treatments. The estimated CRISPR efficiency in schistosome eggs (LE) from CRISPR/*As*Cas12a (green dots in panels g and h, *n* = 8) was at least 1.5% in most samples. There were no statistically significant differences in CRISPR efficiency among the KI_Cas9 vs. KI_Cas12a (panel h) from Sanger sequencing along with TIDE analysis.

**Figure 3 ijms-23-00631-f003:**
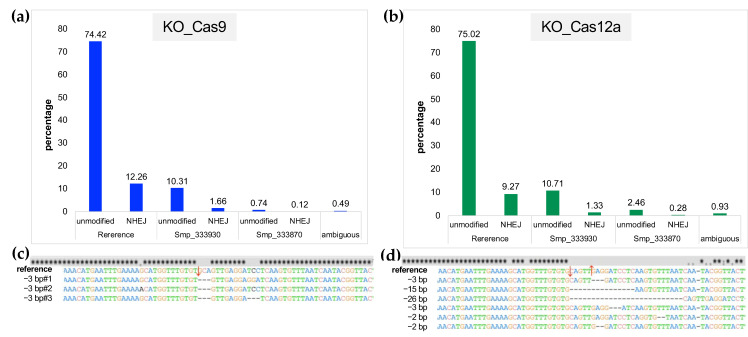
NHEJ assessment from *S. mansoni* eggs at the target *ω*1 gene following delivery of *Sp*Cas9 and *As*Cas12a RNP complexes by electroporation: (**a**–**d**) CRISPResso2 analysis of the Illumina sequence reads revealed nonhomologous end-joining (NHEJ) mutations at targeted loci in schistosome eggs. Allele-specific editing outcomes of CRISPR-*Sp*Cas9 and -*As*Cas12a gene editing in schistosome multiple copies of *ω*1 Smp_334170.2, Smp_334170.1, Smp_334240 (100% gene sequence identical; Appendix A). Mutagenesis frequencies for all loci including Smp_333930 and Smp_333870 were included in CRISPResso2 analysis (Appendix A). The results were from NGS libraries of pooled genomic DNA from seven biological replicates of KO_Cas9 (panels **a**,**c**) and KO_Cas12a (panels **b**,**d**). The sequence reads assigned to reference genes, Smp_333930 and Smp_333870 loci, by the CRISPresso2 to achieve accurate quantification of programmed mutation of the multiple copies of *ω*1 from *Sp*Cas9 and *As*Cas12a; (**a**) the bar chart shows the assignment of each read to the wild type (unmodified) and NHEJ with 12.26%, 1.66%, and 0.12% (total 14.04%) from KO_Cas9 on references and other two loci, respectively; (**b**) shows the assigned NHEJ with 9.27%, 1.33%, and 0.28% (total 10.88%) from KO_Cas12a on references and two highly identical copies of *ω*1; (**c**,**d**) the majority of NHEJ indels from both nucleases were nucleotide substitutions followed by gene deletions; (**c**) Mutations were induced in all copies of *ω*1; however, most editing was seen on the reference target site. Only deletions were revealed in this experiment, 3 nt deletion downstream from expected cut side (red arrow) was the majority of NHEJ mediated from *Sp*Cas9. The larger deletion from 2 to 26 nt at downstream of expected sticky end cut sites (red arrows) were revealed from NHEJ. CRISPResso2 interpreted ≤1.0% of the reads as ambiguous. Ambiguous alignments: 0.49% and 0.93% that could not be attributed uniquely to one of these loci are shown in the right-hand side bar of each graph.

**Figure 4 ijms-23-00631-f004:**
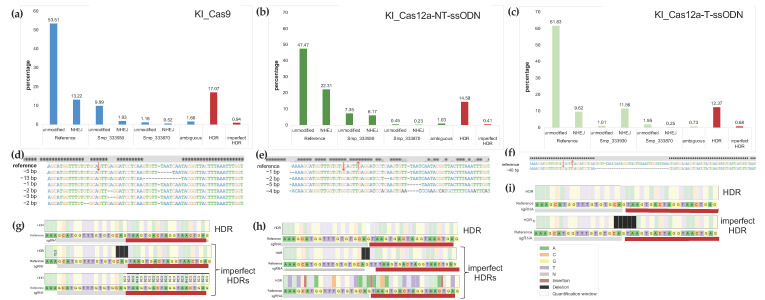
HDR and induction of NHEJ at the *ω*1 target in the presence of donor ssODN templates: (**a**–**c**) show that the percentage of NHEJ from schistosome eggs that introduced RNP along with ssODN template were higher in the presence of ssODN donor templates than in their absence. There were totals of 15.67% (13.22 + 1.16 + 0.52), 28.71% (22.31 + 6.17 + 0.23), and 21.43% (9.62 + 11.56 + 0.25) editing efficiency in the KI_Cas9, KI_Cas12a_NT-ssODN, and KI-Cas12a_T-ssODN groups, respectively (blue or green bars in each graph); (**d**–**f**) Most mutations were substitutions and deletions located downstream of the programmed double-stranded breaks (DSB), with the DSBs indicated by red arrows; (**g**–**i**) A variety of sizes of base deletions from 1 to 15 bp in length were seen in the KI_Cas9, KI_Cas12a_NT-ssODN groups while large deletions, up to 40 bp, were evident as major deletions detected in the KI_Cas12a_T-ssODN group. The percentage of perfect HDR in the three experimental groups, KI_Cas9, KI_Cas12a_NT-ssODN, and KI_Cas12a_T-ssODN were 17.07%, 14.58%, and 12.37%, respectively (red bars). In addition, imperfect HDRs occurred in all KI treatment groups, likely as the result of interruption of 5′ end HDR gene repair by competing NHEJ activity with the result that deletion of 2–5 nt were seen along with donor transgene insertion (black bars in panels **g**,**h**,**i**); the frequency of this event was 0.94, 0.41, and 0.68% among the KI_Cas9, KI_Cas12a_NT-ssODN, and KI_Cas12a_T-ssODN groups.

**Figure 5 ijms-23-00631-f005:**
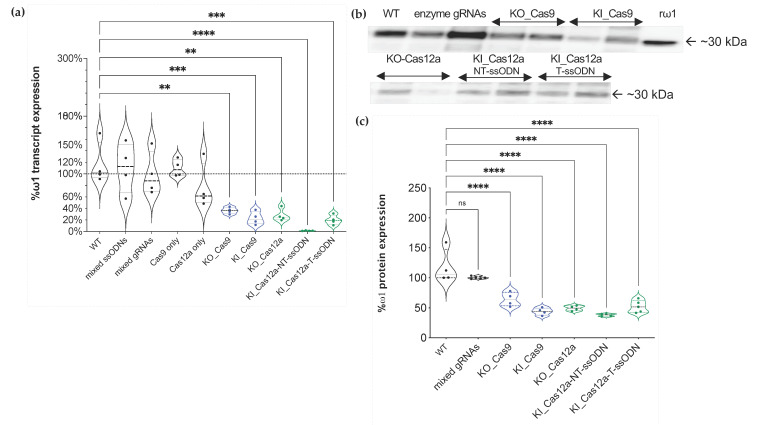
Reduction of the expression of *ω*1 as assessed by quantitative real-time PCR and immunoblot analysis; (**a**) Reduction of *ω*1 transcript levels from KO_Cas9 (blue dots), KI_Cas9 (blue dots), KO_Cas12a (green dots), KI_Cas12a_NT-ssODN (green dots), and KI_Cas12a_T-ssODN eggs (green dots) compared with control groups (black dots) after normalization with wild type (WT) egg RNA (100% as discontinued line). The means represent four independent biological replicates; (**b**,**c**) Soluble egg antigen; SEA (in duplicates) was sized in SDS-PAGE gels, transferred to PVDF, and probed with anti-recombinant omega-1 (r*ω*1) rabbit IgG. Levels of *ω*1 protein, migrating at ~30 kDa, were reduced in all treatment groups. The percentage of *ω*1 expression in both KO and KI exper-imental groups were compared to *ω*1 levels in WT SEA (100%) (one-way ANOVA, multiple comparison, **, *p* ≤ 0.01, ***, *p* ≤ 0.001, ****, *p* ≤ 0.0001; ‘ns’ indicates not statistically significant).

## Data Availability

Source data are available from the corresponding authors.

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
