# Peer review of "RNA-Guided AsCas12a- and SpCas9-Catalyzed Knockout and Homology Directed Repair of the Omega-1 Locus of the Human Blood Fluke, Schistosoma mansoni"

_ijms, 2022, doi:10.3390/ijms23020631_

Round 1

Reviewer 1 Report

In this manuscript, authors compared the efficiency of the RNA-guided AsCas12a nuclease with SpCas9 from Streptococcus pyogenes.  AsCas12a was found to be more efficient than SpCas9 for gene knockout as determined by. Knockout efficiency of both nucleases in the presence of ssODN templates. Although AsCas12a induced fewer mutations per genome than SpCas9, the phenotypic impact on transcription and expression of omega-1 was similar for both nucleases.  

The results of this comparison might inform decisions for functional genomics manipulations aiming to enhance the efficiency of homology-directed repair (HDR) targeting genes.

The data supports the claim made in the manuscript. I am happy to recommend the publication of this manuscript in the present form.

Author Response

Response to reviewer 1:

We are grateful to the reviewer to have reviewed our study and for their recommendation for publication. We have corrected several typographical errors and provided more clarifications, using track changes.

Reviewer 2 Report

Understanding functional genomics is critical, especially among neglected tropical diseases. The authors provide essential tools for that purpose by developing the CRISPR methodology for functional knockout and knockin of the genes. I highly recommend publishing this paper for publication.

  1. Authors should include details regarding the off-target deletions post CRISPR knockout. How were the ssODNs were selected and ranked should be explained. 

Author Response

Response to reviewer 2:

We are grateful to the reviewer for her/his comments on our study and for the recommendation for publication with minor revision.

We used track changes for correction at several typographical errors. Also, to provide more clarification, we have added the text in the Materials & Methods section dealing with rationale and selection of the specific ssODNs.

In addition, we provide below point-by-point responses for the reviewer’s queries on off-target phenomena, deletions and ssODN selection/ranking. (We assume the reviewer was referring to as CRISPR/Cas associated mutations that were apparent at the other omega-1 gene copies.)

R2-1: Authors should include details regarding the off-target deletions post CRISPR knockout.

  • During our analysis assisted by CRISPResso2, several percentage points of base editing were apparent on the conserved sequence motif in the non-reference copies of omega-1. The genome includes four copies of omega-1, three of which appear to be actively transcribed. As the single gRNA used in this study was able to target all four omega-1 gene copies --- the sequence of guide RNA and protospacer region (targeted by the gRNA) of the four copies of omega-1 are100% identical. This gRNA would be able to edit in all omega-1 gene loci, in addition to the on-target reference copy (2) of omega-1. 
  • Next, other potential off-targets (beyond the non-reference copies of omega-1) were not investigated in this study. That kind of investigation was beyond the ambit of the hypothesis of this study, which was a comparison of the performance of the two Cas nucleases (cas9 and Cas12a) at a model schistosome gene locus (omega-1). Nonetheless, we appreciate the reviewer’s recommendation to investigate off-targeting phenomena, and we will aim to address the issue in a future study.

R2-2: How were the ssODNs were selected and ranked should be explained.

  • The ssODN for Cas9 contained 24 nt of 6 stop codons cassette and non-target strand sequences of either 5’ or 3’ end at the double stranded break (DSB) were used as homology arms (HA). The ssODNs for Cas12a-NT-ssODN contained 24 nt of 6 stop codon cassette and HA complementary to T strand sequences aiming to repair at 23 nt from 5’PAM of Cas12a catalyzed DSB.  The Cas12a-T-ssODN contained the HA reverse complementary to NT stand sequence at 18 nt from 5’PAM and reverse complementary to the 6 stop codons cassette.
  • We hypothesized that the Cas12a-T-ssODN would exhibited reduced HDR efficiency, compared with that of Cas12a-NT-ssODN, as the result of indiscriminate trans-activity of Cas12a activity on single strand DNA (a well described phenomenon in other contexts), since the (now single stranded) 5’PAM and partial crRNA (~18 nt) sequence remained in the donor sequence. However, our investigation of potential in vitro Cas12a/RNP activity failed to identify indiscriminate digestion of the ssODN by activated Cas12a.
  • For clarification, we added details of HA used for the Cas12a donor: lines 551-552 in the Materials and Methods section.

Reviewer 3 Report

The work of Wannaporn Ittiprasert et. al.  entitled " RNA-guided AsCas12a- and SpCas9-catalyzed knockout and homology directed repair of the omega-1 locus of the human blood fluke, Schistosoma mansoni" describes the novel use of ASCas12a nuclease from Acidaminococcus species, in the mutation of the helminth Schistosoma mansoni. The authors compare the gene editing efficiency of AsCas12a with the widely used SpCas9 nuclease. The comparison is in terms of efficiency and precision where prove to be comparable between the two nucleases. The big advantage of the AsCas12a is that can be used on AT rich genome (T rich PAM sites) and without loosing precision or efficiency. They also employ the use of small ssODN nucleotides sequences for HDR that are not cleaved by the nucleases. The work is solid but some key points need to be clarified and most importantly the repair mechanisms used in the cases of KO and KI. 

Also, it is advised that the manuscript  would be proof read for English phrasing since it is quite difficult in parts to understand.

Abstract:

line 20: correct the typo prgrammed

introduction:

line 58:  for complex genome editing like the one in plants or for example in plants

lines 63-66: there is no information as how mutations on S. mansoni are previously created e.g.  delivering the RNP complexes is the only way or there are alternative paths like vector design. Please rewrite or explain the sentence "Programmed CRISPR/Cas editing using RNP-based delivery 
can precede analysis of mutated target regions that rely on cloning and vector construction". 

Results:

line 80: encodes

line 85: missing coma or dot in the location 3,984675

line 86: the genes are four but the positions three. It seems that the first location contains two genes if each gene is around 2.3kb, please clarify.

line 90: there is no Smp_333220 in figure S1b. 

line 93:correct typo strand

line 98: please define that there is another ssODN for the ASCas12a editing

line 100: if PAM is retained in the HDR how do you ensure that is not cleaved after gene repair? This would lead to cycles of cleavage after repair occurs. Does PAM site is altered in the HDR  when using spCas9 .

line 102 figure 1 legend: the description of 1a and 1b is mixed. in addition there are not yellow arrows on figure 1a as described. The authors describe that they did ko and kI assays . However only one set of ssODN are presented. please clarify what ssODN are used in each situation and provide the sequences in figure1. in the case that no ssODN are provided for the ko assays then clarify which repair mechanism is used e.g. NEHJ?. What the underlined part is in the fig1c?

line 119: rewrite the phrase "the ω1 gene activated AsCas12a". AsCas12a nuclease is not activated by ω1 gene. 

line 127: delete the extra 'the'

line 130: 'in vitro' should be in italics

line 133 Figure 2a:   explain what the green box is. 2b: these are drawing and not the original digests. Please include the gels from the restriction digestions.2d: lane 3 is not described.2d&2e-f. please add which primers are used to be easy to follow. 2g-h  do you mean blue dots?

line 165-166: which primers are used for the standard curve? The sentence is unfinished.

line 181: why only the crRNA and not the gRNA was complexed with AsCaa12a

line 195 and line 199. please clarify where 'above' refers to AKA what is pooled earlier in line 195 rather than line 199.

line 205: what does it mean that the reads are ambiquous?

line 211: the way the sentence '
The numbers of indels arising from NHEJ with SpCas9 and with AsCas12a both increased in the presence of a ssODN donor repair template" is written reads like the NHEJ mechanisms works in the presence of ssODN. please rewrite the sentence for clarity.

line 212: do you mean indel instead of NHEJ?

line 240: the (d) is missing 

line 248: mutations created from which repair mechanism?

line 259:  Delete 'the experimental'

line 270: what is contained in the 'mixed gRNA only group'

Discussion

Lines 321 to 323: it is unclear what authors mean in the sentence: 'In some cases, the two sides of the DSB programmed by AsCas12a were repaired by different pathways since mutant alleles that included only the left or only the right side of the transgene were observed'. It is unclear if they mean that sequences that are repaired by both mechanisms were found with NGS or sequences that one part was repaired by HDR and the other by NHEJ. However I cannot see how the later can be detected so please rewrite the sentence for clarity.

Line 334:In vitro in italics

Author Response

Response to reviewer 3:

We are grateful to the reviewer to have reviewed our study and for his/her recommendations and suggestions.

R3-Abstract:

line 20: correct the typo prgrammed

Corrected as ‘programmed’.

R3-Introduction:

line 58:  for complex genome editing like the one in plants or for example in plants

Revised as: ‘These Cas12a features are also attractive for complex genome editing including in plants [12].’

lines 63-66: there is no information as how mutations on S. mansoni are previously created e.g.  delivering the RNP complexes is the only way or there are alternative paths like vector design. Please rewrite or explain the sentence "Programmed CRISPR/Cas editing using RNP-based delivery can precede analysis of mutated target regions that rely on cloning and vector construction". 

Sentence revised: ‘Programmed CRISPR/Cas editing using RNP-based delivery can precede analysis of mutated target regions that rely on cloning and CRISPR vector construction, as addressed in our earlier reports [1, 2].’

R3-Results:

line 80: encodes

Corrected as ‘encodes’.

line 85: missing coma or dot in the location 3,984675

Comma added.

line 86: the genes are four but the positions three. It seems that the first location contains two genes if each gene is around 2.3kb, please clarify.

We apologize for the lack of clarity on the positions for Smp_334170.  We have modified as Smp_334170.1 and Smp_334170.2 as now indicated in the text.

line 90: there is no Smp_333220 in figure S1b.

We have now included Smp_333220 in the alignment, Figure S1b.

line 93: correct typo strand

We checked the spelling of ‘strand’, ‘strands’, and ‘stranded’ throughout the manuscript and corrected any problems.

line 98: please define that there is another ssODN for the ASCas12a editing

Additional information of the ssODNs including their homology arms, has been in the Materials and Methods section, lines 551-552.

line 100: if PAM is retained in the HDR how do you ensure that is not cleaved after gene repair? This would lead to cycles of cleavage after repair occurs. Does PAM site is altered in the HDR  when using spCas9 .

Thank you for the question.  The 5’PAM sequence on the ssODN donor of Cas12a may affect the HDR efficiency due to potential indiscriminate trans-activity of activated Cas12a/RNP.  However, we failed to demonstrate the indiscriminate trans cleavage by activated Cas12a against the ssODN in vitro; the findings of these assays are presented in the Results section, 4.11. 

After permanent donor integration into the target site, retention of 5’PAM/Cas12a and stop codons cassette would not have occurred here because of the expected short biological half-life of crRNA/Cas12a ribonucleoprotein complex within cells of the schistosome egg.  A short term Cas12a activity may occur in the egg cells after HDR; however, after several days following the KI delivery of the CRISPR system by electroporation, there were >12% perfect HDR from CRISPR/Cas12a using either the Cas12a-T-ssODN or Cas12a-NT-ssODN donors.  In future studies, modifications of the DNA donor particularly the PAM site of Cas12a, will be considered with the goal of obviating any possibility of indiscriminate enzymatic cleavage.

For the 3’PAM site of Cas9 on the ssODN donor or after KI into the target site, this event should not affect the HDR because there is 24 nt of stop codon cassette sequence instead of CRISPR/Cas9 target sequence. 

line 102 figure 1 legend: the description of 1a and 1b is mixed. in addition there are not yellow arrows on figure 1a as described. The authors describe that they did ko and kI assays . However only one set of ssODN are presented. please clarify what ssODN are used in each situation and provide the sequences in figure1. in the case that no ssODN are provided for the ko assays then clarify which repair mechanism is used e.g. NEHJ?. What the underlined part is in the fig1c?

The three ssODN sequences shown in figure 1c are as follows: Cas9-ssODN (red color letters), Cas12a-NT-ssODN (light blue color letters) and Cas12a-T-ssODN (glue color letters).  The yellow arrows indicate 5’PAM_cas12a or 3’PAM_cas9 in figure 1b. We have modified the figure legend - as shown in tracked change in the manuscript. 

in the case that no ssODN are provided for the ko assays then clarify which repair mechanism is used e.g. NEHJ?

Correct; non-homologous end joining (NHEJ) is expected to effect the mutation repair without insertion of the ssODN donor template.  In our study, the prediction of NHEJ in the KO-focused assays was analyzed by CRISPResso. The results are provided in figure 3. 

For the KI focused assays, both NHEJ and HDR mechanisms were found to have been employed for omega-1 gene repairs. The CRISPResso analysis results are shown in figure 4.

What the underlined part is in the fig1c?

Figure 1C: the underlined motif indicates the stop codons.  For clarity, an explanation is now included in the figure legend.

line 119: rewrite the phrase "the ω1 gene activated AsCas12a". AsCas12a nuclease is not activated by ω1 gene. 

We apologize for the typographical error.  This has been revised as ‘activated AsCas12a’ as the reviewer recommended.

line 127: delete the extra 'the'

Deleted.

line 130: 'in vitro' should be in italics

Done.

line 133 Figure 2a:   explain what the green box is.

The green box indicates the 24 nt transgene (stop codon cassette). This point has been clarified in the revised legend of the figure (track changes).

The revised Figure legend now reads: ‘(a) The location of primers used for 5’- and 3’- transgene (purple lines) knock-in PCR (5’int-F with 24nt-6stp-R and 24nt-6stp-F with 3’int-R) and internal positive control (control-F and -R).  The primers for next generation sequencing indicate as NGS-F and NGS-R.  All primers are located outside of the homology arms; HAL and HAR (purple boxes) sequences to avoid false positive amplification from DNA donor. The 24 nt-6stp-F and -R primers are highly specific for the stop codon transgene and should not anneal to the wild type schistosome gene. The green box shows the position of the 24 nt six stop codon transgene’.

2b: these are drawing and not the original digests. Please include the gels from the restriction digestions.

The figures in 2b are original electropherograms from the BioAnalyzer electrophoresis. They are not agarose gels/ electrophoresis, as explained in the Materials and Methods and Result sections.  Electropherogram based analysis using the Agilent BioAnalyzer provides higher sensitivity and precision for fragmented DNAs.  We used the BioAnalyzer approach instead of conventional agarose analysis because we anticipated the need for higher resolution (than agarose gel electrophoresis paired with ethidium bromide staining) to detect of any gel fragments resulting from nuclease activity.

2d: lane 3 is not described.

Lane 3 describes as ‘The crRNA/AsCas12a RNP (~40 nt in lanes 3-9) containing activated …’ on the line 264-265.

2d&2e-f. please add which primers are used to be easy to follow

The primers are listed in supplementary table S1.  As the reviewer recommended, the primer names for 5’ and 3’ integration focused PCRs are included now are included in the figure legend.

2g-h  do you mean blue dots?

The code for dot colors has been clarified as in the figure legend.

line 165-166: which primers are used for the standard curve? The sentence is unfinished.

We revised the first two sentences of the paragraph to clarify the issue, as follows:

‘An external standard curve design was employed to estimate transgene copy number in genomics DNA of the schistosome eggs (Fig. 2e and 2f). The standard curve is based on simple linear regression (y = -4.0964ln[10] + 44.821, R² = 0.9923) established with logarithm-10 transformed initial DNA input, plasmid pCR4, which encodes the six stop codon cassette [1], as the dependent and the Ct value from the qPCRs as the independent variable (not shown)’. 

The qPCR of transgene knock-in using plasmid pCR4 as the template and the 5’ or 3’ integration PCR primers for the standard curve.

line 181: why only the crRNA and not the gRNA was complexed with AsCaa12a

Because there is no tracrRNA in the process of complexing of crRNA with the Cas12a protein for programmed DNA cleavage by Cas12a.

line 195 and line 199. please clarify where 'above' refers to AKA what is pooled earlier in line 195 rather than line 199.

‘Above’ have been deleted.  The sentence ‘…pooled-genomic DNA samples after TIDE analysis by target amplicon reads by MiSeq 2´250 bp…’ was modified using track changes.

line 205: what does it mean that the reads are ambiquous?

The outcome for ‘ambiguous’ resulting from CRISPREsso2 were the reads with two or more possible or if it was unclear whether those reads could align to any one of the reference sequences, ever though those reads were from fully paired forward and reverse reads of NGS data.  Also, those reads were not low quality reads given that low quality and/unpaired reads were eliminated before CRISPResso analysis.

line 211: the way the sentence '
The numbers of indels arising from NHEJ with SpCas9 and with AsCas12a both increased in the presence of a ssODN donor repair template" is written reads like the NHEJ mechanisms works in the presence of ssODN. please rewrite the sentence for clarity.

The sentence has been revised as follows:

‘The numbers of indels arising from NHEJ with SpCas9 and with AsCas12a both increased in the presence of a ssODN donor repair template in the case of HDR did not dominant (expected) to repair the target site in the parasite eggs’. 

line 212: do you mean indel instead of NHEJ?

Thank you; we meant NHEJ.  The extra ‘indel’ have been removed.

line 240: the (d) is missing 

Corrected.

line 248: mutations created from which repair mechanism?

By NHEJ in this case.

line 259:  Delete 'the experimental'

Deleted.

line 270: what is contained in the 'mixed gRNA only group'

It was a negative control group; the group was treated by electroporation with a mixture of both guide RNAs (of Cas9 and Cas12a) but in the absence of the Cas nucleases.

Discussion

Lines 321 to 323: it is unclear what authors mean in the sentence: 'In some cases, the two sides of the DSB programmed by AsCas12a were repaired by different pathways since mutant alleles that included only the left or only the right side of the transgene were observed'. It is unclear if they mean that sequences that are repaired by both mechanisms were found with NGS or sequences that one part was repaired by HDR and the other by NHEJ. However I cannot see how the later can be detected so please rewrite the sentence for clarity.

Sentence revised as follows:

‘In some cases, the two sides of the DSB programmed by AsCas12a were repaired by different pathways, e.g., one side repaired by HDR and the other by NEHEJ, since mutant alleles that included only the left or only the right side of the transgene were observed [45]’. 

This situation has been described by others previously – please see Ref. 45, which we cited here in support of our statement.

Line 334: In vitro in italics

Italicized as recommended.
